# Effects of Pesticides on Longevity and Bioenergetics in Invertebrates—The Impact of Polyphenolic Metabolites

**DOI:** 10.3390/ijms222413478

**Published:** 2021-12-15

**Authors:** Fabian Schmitt, Lukas Babylon, Fabian Dieter, Gunter P. Eckert

**Affiliations:** Biomedical Research Center Seltersberg (BFS), Laboratory for Nutrition in Prevention and Therapy, Institute of Nutritional Sciences, Justus Liebig University Giessen, Schubertstrasse 81, 35392 Giessen, Germany; Fabian.Schmitt@ernaehrung.uni-giessen.de (F.S.); Luksa.Babylon@ernaehrung.uni-giessen.de (L.B.); Fabian.Dieter@ernaehrung.uni-giessen.de (F.D.)

**Keywords:** *Caenorhabditis elegans*, pesticides, fluopyram, glyphosate, pyraclostrobin, mitochondria

## Abstract

Environmentally hazardous substances such as pesticides are gaining increasing interest in agricultural and nutritional research. This study aims to investigate the impact of these compounds on the healthspan and mitochondrial functions in an invertebrate in vivo model and in vitro in SH-SY5Y neuroblastoma cells, and to investigate the potential of polyphenolic metabolites to compensate for potential impacts. Wild-type nematodes (*Caenorhabditis elegans,* N2) were treated with pesticides such as pyraclostrobin (Pyr), glyphosate (Gly), or fluopyram (Fluo). The lifespans of the nematodes under heat stress conditions (37 °C) were determined, and the chemotaxis was assayed. Energetic metabolites, including adenosine triphosphate (ATP), lactate, and pyruvate, were analyzed in lysates of nematodes and cells. Genetic expression patterns of several genes associated with lifespan determination and mitochondrial parameters were assessed via qRT-PCR. After incubation with environmentally hazardous substances, nematodes were incubated with a pre-fermented polyphenol mixture (Rechtsregulat^®^Bio, RR) or protocatechuic acid (PCA) to determine heat stress resistance. Treatment with Pyr, Glyph and Fluo leads to dose-dependently decreased heat stress resistance, which was significantly improved by RR and PCA. The chemotaxes of the nematodes were not affected by pesticides. ATP levels were not significantly altered by the pesticides, except for Pyr, which increased ATP levels after 48 h leads. The gene expression of healthspan and mitochondria-associated genes were diversely affected by the pesticides, while Pyr led to an overall decrease of mRNA levels. Over time, the treatment of nematodes leads to a recovery of the nematodes on the mitochondrial level but not on stress resistance on gene expression. Fermented extracts of fruits and vegetables and phenolic metabolites such as PCA seem to have the potential to recover the vitality of *C. elegans* after damage caused by pesticides.

## 1. Introduction

Environmental chemicals with hazardous potential are widely used in agricultural industries. Commonly known under the term “pesticides”, this word covers a wide range of compounds that include insecticides, fungicides, herbicides, rodenticides, molluscicides, nematicides, and others [1,2]. They not only reduce crop losses due to pests, but they also improve the quality and yield of produce [3]. Even in terms of optical appearance, pesticides harbor a wide range of benefits [4]. However, besides the potential agricultural advantages of pesticides, a broad spectrum of intensive studies over the years has declared these synthesized compounds to be hazardous to health [5,6,7]. Extensive research over the years has led to the realization that the harmful effects of pesticides include genotoxicity [8,9], teratogenicity, embryotoxicity [10], and perhaps carcinogenicity [11].

There are major groups of pesticides that are conventionally used in today’s agricultural elimination of crop and other harmful diseases, and those highlighted include the strobilurins, phosphonates, and pyridinyl ethyl benzamides, which belong to the category of fungicides and herbicides. The group of strobilurins originates as a natural product, and was first isolated from the mycelium of Basidomycete *Strobilurus tencellus* [12]. Synthetically produced strobilurins, such as pyraclostrobin (Figure 1, Table 1), act as fungicides and inhibit mitochondrial respiration by binding to the ubihydrochinone oxidation center of complex III, the cytochrome c–oxidoreductase [13]. This inhibition blocks electron transferring through the electron transfer chain [14,15]. Another group of fungicides are the pyridinyl ethyl benzamides. A commonly used agent of this group is fluopyram (Figure 1, Table 1) [16]. The mode of action of this substance has been shown to inhibit complex II of the respiratory chain, enzyme succinate dehydrogenase [17,18,19]. This fungicide shows biological activity against all stages of fungal growth [16,20].The inhibiting effect on succinate dehydrogenase of fluopyram blocks energy production as well as the production of precursor substances, which are used for the synthesis of cellular compounds such as amino acids [16]. The phosphonates are well known for their famous agent, glyphosate (Figure 1, Table 1) [21,22]. This non-selective herbicide was first tested, or at least patented, for use as an herbicide only in 1970 [23]. Glyphosate is highly effective at inhibiting the enzyme 5-enolpyruvyl-shikimate-3-phosphate synthase (EPSPS) [24]. This enzyme belongs to the shikimate pathway, thus its action results in the biosynthesis of aromatic amino acids in plants [25]. As a major compound of the commercial product Roundup Ready*^®^*, it is one of the most discussed and tested pesticides available on the market [26]. Many studies over the years have examined glyphosate, as well as its commercially available mixture, although did not clarify the potentially harmful effects on cancer progression [27,28,29].

As previously mentioned, one of the main targets of pesticides are mitochondria. As a central element, mitochondria harbor the TCA-cycle as well as oxidative phosphorylation (OXPHOS) to generate the universal energy source, adenosine triphosphate (ATP) [30,31,32]. Dysfunctions of this elementary process are associated with many diseases and the advancement of aging [33]. Even if pesticides were only intended to affect plants, fungi, or insects, there is now increasing evidence that they also affect the mitochondria of mammals and can lead to changes in energy metabolism, which, in turn, cause harmful diseases and neurotoxic effects [34,35,36,37].

The roundworm *Caenorhabditis elegans* (*C. elegans*) is a popular model organism that is used to study the effects of toxic substances [38] and mitochondrial dysfunction [39]. The invertebrate *C. elegans* proves to be a good model to understand the role of mitochondria, and its usage provides knowledge on a sub-cellular, tissue-specific, and organismal level because of the link between the mitochondria and the lifespan of the nematodes [40].

Here, we hypothesized the impact of three selected and commonly used pesticides on stress resistance, chemotaxis, and mitochondrial bioenergetics and investigated the effects by co-treatment with pre-fermented polyphenols. Therefore, we used methods including a thermotolerance survival assay, a behavior assay, quantification of energy metabolites, and the gene expression pattern of several genes related to longevity and mitochondrial parameters.

**Table 1 ijms-22-13478-t001:** Pesticides used in this study with their target enzyme.

Pesticide	Target	Acute Toxicity	Literature
Glyphosate	EPSPS	LD50 Oral: Rat—4.873 mg/kg LD50 Dermal: Rabbit—2.000 mg/kg	[41,42,43,44,45,46,47]
Fluopyram	Succinate dehydrogenase (complex II)	LD50 Oral: Rat—>2.000 mg/kg LD50 Dermal Rat—>2.000 mg/kg	[17,19,48,49]
Pyraclostrobin	cytochrome c–oxidoreductase (complex III)	LD50 Oral: Rat—>5.000 mg/kg LC50 Inhalation: Rat—4 h—0.31—1.07 mg/L LD50 Dermal: Rat—>2.000 mg/kg	[50,51,52,53,54,55]

## 2. Material and Methods

### 2.1. Chemicals

The chemicals used in this study were of the highest available purity and standard from Merck (Darmstadt, Germany).

### 2.2. Cells

In this study, SH-SY5Y cells were used. The cells were grown in 250 mL Greiner flasks with Dulbecco’s modified Eagle medium (DMEM) (Gibco, Thermo Scientific, Waltham, MA, USA), supplemented with 10% (*v*/*v*) fetal bovine serum (FBS), 1% MEM-vitamins, pyruvate, and nonessential amino acids and antibiotics (penicillin, streptomycin). For selectivity, 3 µg/mL hygromycin B was added to the medium. When cell growth reached a confluency of 70–80%, cells were transferred to a new culture flask.

For the experiments, cells were harvested from Greiner flasks, counted using a Neubauer Chamber, and were diluted to yield a cell suspension of 10^6^ cells/mL. Cells were then sown into 96-well plates (ATP, Autophagy and ROS assays, 2 × 10^4^ cells/well). Cells were seeded in reduced DMEM (2% FBS and other supplements identical to cultivating medium) and were allowed to attach to the bottom of the wells for 48 h before being exposed to 1 mM to 10 nM pyraclostrobin. Pyraclostrobin was prepared in EtOH. Its final concentration in all experiments ranged from 0.1% to 1%.

### 2.3. Nematode and Bacterial Strain

Wild-type nematode strain N2 was obtained from the Caenorhabditis Genetics Center (University of Minnesota, Minneapolis, MN, USA). Nematodes were maintained on a nematode growth medium (NGM) agar plates seeded with the bacterial *E. coli* strain OP50. According to standard protocols, the seeded plates were stored at 20 °C [56]. Synchronous populations were generated for all experiments by using a standard bleaching protocol [57].

### 2.4. Cultivation and Treatment

Post-bleaching generated larvae were washed twice in an M9 buffer, and the number of larvae in 10 µL was adjusted to 10 larvae. Afterward, the synchronized larvae were raised in cell culture flasks (Sarstedt, Nümbrecht, Germany) in either 1000 or 5000 nematodes, depending on the experiments. Furthermore, OP50-NGM was added to the flasks as a standardized source of food. The larvae were maintained under shaking at 20 °C until they reach young adulthood within 3 days.

The chemicals were dissolved in advance. Pyraclostrobin and fluopyram were first fully dissolved in 100% ethanol and were then diluted in M9 to create the required concentrations. Glyphosate was fully dissolved in H_2_O bidest. Protocatechuic acid was dissolved in 10% ethanol in M9 buffer. The final concentration of ethanol after treatment was 1%. As a control group in all the experiments, M9 was used with an amount of 1% ethanol or as neat.

After reaching adulthood (48 h prior to the experiment), the chemicals were added to the flasks.

### 2.5. Heat Stress Survival Assay

After 48 h of incubation with the mentioned effectors, the time until death was determined using a microplate thermotolerance assay [58]. During preparation, the nematodes were washed out of the flasks with M9 buffer into 15 mL tubes, followed by three additional washing steps. Each well of a black 384-well low-volume microtiter plate (Greiner Bio-One, Frickenhausen, Germany) was prefilled with 6.5 µL M9 buffer/Tween^®^20 (1% *v*/*v*). In the following step, one nematode in 1 µL M9 buffer was transferred and immersed in the well under a stereomicroscope (Breukhoven Microscope Systems, Essebaan, The Netherlands). SYTOX™ Green (Life Technologies, Karlsruhe, Germany), in a final concentration of 1 µM, was added to reach a final volume of 15 µL in the well. SYTOX™ Green creates a fluorescent signal after binding to DNA. The plates were sealed with a Rotilab sealing film (Greiner Bio-One, Frickenhausen, Germany). At a temperature of 37 °C, heat shock was applied, and the fluorescence was measured with a ClarioStar Plate Reader (BMG, Ortenberg, Germany) every 30 min over the course of 17 h. The excitation was set at 485 nm, and the emission was detected at 538 nm.

### 2.6. Chemotaxis Assay

Chemotaxis was assessed using a previously published method [59]. The agar plates were divided into four quadrants. Sodium acid (0.5 M) was mixed in the same parts with ethanol (95%) as a control or diacetyl (0.5%) as an attractant. Either 2 µL of the control or attractant solution was added to the center of two opposite quadrants at the same distance to the middle of the plate. The nematodes were washed out of the flasks and separated from larvae. Approximately 150 animals were placed in the plate’s center. After 1 h, each quadrant was counted, and a chemotaxis index was calculated: Chemotaxis index = (# worms in both test quadrants (−) # worms in both control quadrants)/(total # of scored worms). The calculated chemotaxis ranks between −1.0 and a +1.0. A +1.0 score indicates maximal attraction toward the target and represents 100% of the worms arriving in the quadrants containing the chemical target. An index of −1.0 is evidence of maximal repulsion.

### 2.7. Nematode Homogenization

Additionally, 5000 synchronized nematodes were thoroughly washed out of the flasks, shock frozen in liquid nitrogen, and stored until use at −80 °C. The samples were boiled for 15 min before sonication (Cycle 1, Amplitude 100%) to denature the degrading proteins. After centrifugation at 15,000 g for 10 min, supernatants were collected. ATP, lactate, pyruvate, and protein content was assessed out of these aliquots and were stored between experiments at −80 °C.

### 2.8. ATP Measurement

Intracellular ATP levels were determined using the ATPlite luminescence assay system (Perkin Elmer, Waltham, MA, USA). Luminescence was measured in triplicate following the manufacturers’ guidelines with a ClarioStar Plate Reader (BMG, Ortenberg, Germany). Aliquots were stored at −80 °C for the determination of protein content and other metabolites.

### 2.9. Colorimetric Assessment of Lactate and Pyruvate Content

Frozen homogenate samples were slowly thawed until reaching room temperature. Concentrations of lactate and pyruvate were detected by changes in the NADH content using two colorimetric assay kits from Sigma-Aldrich following the manufacturer’s guidelines for either lactate or pyruvate (Sigma-Aldrich, St. Louis, MO, USA) using a ClarioStar Plate Reader (BMG, Ortenberg, Germany).

### 2.10. Protein Quantification

Protein contents were assessed according to the Pierce™ BCA Protein Assay Kit (Thermo Fisher Scientific, Waltham, MA, USA). Bovine serum albumin was used as a standard.

### 2.11. MTT Cell Viability Test

To perform the MTT cell viability test, 10,000 cells per well were seeded in a 96-well plate. On the following day, the cells were incubated with the respective agents for 24 h. Two hours before the end of the incubation period, 20 μL of MTT solution (37 °C) was pipetted into all wells, and the plate was then placed back into the incubator. At the end of the incubation period, the cell medium with the MTT solution was carefully aspirated with a Pasteur pipette. Following this, 100 μL DMSO was added to dissolve the crystals. To ensure that all crystals were dissolved, the 96-well plate was placed on a shaker for 5 min at room temperature. The absorbance of the dissolved formazan was measured at a wavelength of 570 nm using a ClarioStar Plate Reader (BMG, Ortenberg, Germany).

### 2.12. Quantitative Real-Time PCR

Using the RNeasy Mini Kit (Qiagen, Hilden, Germany), total RNA was isolated according to the manufacturer’s guidelines after homogenization of the nematodes using a Balch Homogenizer with 10 µm clearance. The concentration of the RNA content was quantified by measuring the absorbance at 260 and 280 nm using a NanoDrop™ 2000c spectrophotometer (Thermo Fisher Scientific, Watham, MA, USA). RNA purity was assessed with the ratio absorbance at 260/280 and 260/230 nm, separately. After this, the samples were treated with TURBO DNA-free Kit™ (Thermo Fisher Scientific, Watham, MA, USA) to remove residual genomic DNA. According to the manufacturer’s guideline, complementary DNA was synthesized from 1 µg total RNA using the iScript cDNA Synthesis Kit (Bio-Rad, Munich, Germany). The samples were stored at −80 °C until used. The qRT-PCR was conducted using a CfX 96 Connect™ system (Bio-Rad, Munich, Germany). The used primers were purchased from BioMers (Ulm, Germany). In Table 2, all oligonucleotide primer sequences, primer concentrations, and product sizes are listed. The cDNA samples were diluted in a 1:10 distribution with RNase-free water (Qiagen, Hilden, Germany), and the samples were measured in triplicates. PCR cycling conditions were at an initial denaturation at 95 °C for 3 min, followed by 45 cycles of 95 °C for 10 s, 58 °C for 45 s (*aak-2* was at an expectation at 62 °C), and extension at 72 °C for 29 s. Gene expression levels were analyzed by applying the (2ΔΔC_q_) method using Bio-Rad CfX manager software and were normalized to the expression levels of *ama-1* and *act-2*.

### 2.13. Statistics

Unless otherwise stated, values are presented as mean ± standard error of means (SEM). Statistical analyses were performed by applying a one-way analysis of variance (ANOVA) with Tukey’s multiple comparison post hoc test (Prism 9.1 GraphPad Software, San Diego, CA, USA). Results with *p* values * *p* < 0.05, ** *p* < 0.01, *** *p* < 0.001, and **** *p* < 0.0001 were considered statistically significant.

## 3. Results

### 3.1. Pesticides Reduce the Survival under Heat Stress at 37 °C

To investigate the potentially damaging effect of the investigated pesticides, it was first necessary to evaluate a range of concentrations that led to a decreased heat-stress resistance. After exposure to several concentrations of glyphosate (Figure 2a,b; 3, 1, 0.1, 0.01, and 0.001 mM), fluopyram (Figure 2c,d; 100, 10, 5, 2.5, and 1 µM) and pyraclostrobin (Figure 2e,f; 1, 100, 10, 5, and 1 µM), the heat stress of the nematodes showed a dose-dependent and significant decline. Based on these results, we used relevant concentrations of the pesticides for further experiments. For glyphosate, 0.001 mM was the chosen concentration because it led to a decrease of 19% in mean lifespan. In the case of fluopyram and pyraclostrobin, the concentrations 2.5 and 5 µM were chosen because they were the first concentrations that led to a significant decline in the mean survival in the heat stress assay.

### 3.2. Chemotaxis Is Not Altered after Exposure to Pesticides

The analysis of the chemotactic ability of the nematodes to locate food was not decreased by the exposure of the nematodes to the pesticides. None of the investigated pesticides (glyphosate (Figure 3a; 0.0028482 ± 0.1422), fluopyram (Figure 3b; −0.02537 ± 0.1399) and pyraclostrobin (Figure 3c; −0.04742 ± 0.1385)) showed any significant decrease of chemotaxis.

### 3.3. Energy Metabolites Were Altered in Different Ways by Pesticides

Next, the impact on the mitochondrial energy metabolism was investigated. The major energetic metabolite ATP was significantly increased by 56% after the exposure to pyraclostrobin (Figure 4a). Fluopyram and glyphosate increased ATP levels by 17% and 18%, respectively. The other energetic metabolites show similarly increased levels. Comparable to the ATP levels, pyraclostrobin elevated the levels of pyruvate and lactate (Figure 4b,c) but did not alter the lactate-to-pyruvate ratio (L/P ratio) (Figure 4d). In contrast, glyphosate and fluopyram tended to increase pyruvate levels but did not increase lactate levels without reaching significance. Fluopyram tended to decrease the L/P ratio but was not significant (Figure 4d).

To evaluate the toxic effect of pyraclostrobin in other models, we used SH-SY5Y cells. As described above, we observed the mitochondrial energy metabolism, but we also investigated cell viability with an MTT assay. In this experiment, we observed a range of concentrations between 1 mM and 10 nM. The cells were treated with pyraclostrobin for 24 h, which displayed a short-term effect on the cells. For ATP levels, we observed an almost overall significant decrease (Figure 5a). The chosen concentrations (5 µM) during the nematode experiments led to a significant reduction of ATP levels by 64.53%. Higher concentrations resulted in a decline of ATP by almost 90%. Regarding cell viability, Pyr 5 µM was not found to lead to alteration. In higher concentrations, similar to ATP, cell viability was significantly reduced (Figure 5b).

### 3.4. Effects of Pesticides on mRNA Expression

The analysis of the expression of mRNA by qRT-PCR reveals a significant decrease in mRNA levels of all target genes (Table 3). Interestingly, we observed a decrease of the longevity-related marker genes *sir-2.1* (chromosome organization, determination of adult lifespan, intrinsic apoptotic signaling pathway), *skn-1* (endoderm development, endoplasmatic reticulum unfolded protein response, multicellular organism development)*,* and *daf-16* (defense response to other organisms, regulation of dauer larval development, regulation of primary metabolic process). Pyraclostrobin treatment resulted in a sharp drop of values compared to the control. Fluopyram and glyphosate also caused a numerical decrease in mRNA levels in these markers, but changes were not significant. Furthermore, the mRNA level of *atfs-1*, responsible for the mitochondrial-unfolded protein response, was lowered by exposure to pesticides. Besides a significantly decreased level of *atp-2* by pyraclostrobin, which encodes for a subunit of ATP synthase, fluopyram managed to significantly increase the mRNA levels of this gene compared to untreated nematodes.

### 3.5. Heat Stress Resistance Is Restored after Treatment with Phenolic Metabolites

The previously mentioned toxic damages on heat stress resistance could be reversed after co-treatment with either Rechtsregulat^®^Bio (RR) or protocatechuic acid (PCA). RR represents a fermented fruit and vegetable drink, rich in polyphenols, that led to an extension of the lifespan of mice and *C. elegans* in our previous study [60]. The fermentation process led to a change of the bioactive compounds in the product. Phenolic compounds are converted into molecules with an increased biological value [61]. PCA is a well-established phenolic acid, representing a metabolite of polyphenols including quercetin [62,63], and it is a major compound of RR. PCA also prolonged the lifetime of nematodes as well as imparting reverse effects after exposure to paraquat. In our present study, we co-treated the worms with pesticides and RR or PCA. RR and PCA compensated for the effects of glyphosate (Figure 6a,b) on heat stress resistances, which were reversed. The exposure to fluopyram (Figure 6c,d) or pyraclostrobin (Figure 6e,f) and their reducing effect on heat stress resistance were also restored after treatment with RR or PCA.

## 4. Discussion

A popular model to study pesticides is the nematode *C. elegans* [64]. Since the nematodes were isolated from the soil, they came to contact with various environmental contaminants used in the agricultural industry [65,66,67]. In their original use, pesticides were utilized in preparations to repel, destroy or control pests. Until today, they have proven to be an essential tool in agriculture and public health. Despite their usefulness, the molecular targets of pesticides are often the same, including in humans. Herbicides and fungicides, theoretically, should not affect mammals as targets, but several pesticides have been demonstrated to influence, for instance, the mammalian brain [68]. In this study, we hypothesized a possible influence of glyphosate, a widely used herbicide, and the fungicides fluopyram and pyraclostrobin on stress resistance and mitochondrial parameters. *C. elegans* provides an optimal organism to study the mechanism after pesticide treatment, because of changes in behavior, propagation, and growth when exposed to some metals and organic compounds [69]. This fact makes the nematode a suitable model to assess adverse effects on aquatic and soil organisms. The use of *C. elegans* for toxicological studies offers many advantages. *C. elegans* shows considerable similarities to mammals in terms of biochemistry and genomics [70]. Alongside toxicants that are harmful to mammals, anthelmintic drugs and nematicides are studied in this model organism [71,72,73,74].

Stress resistance is decreased by the toxic influence of pesticides. Throughout the literature, pesticides were commonly observed by the administration of mixtures containing the active compound. In agricultural use, manufacturing companies provide their products as complex preparations, which also consist of other compounds such as stabilizers with possible harmful effects. Authors of other studies have already shown synergistic effects between glyphosate and other ingredients of the commercially available product Roundup Ready*^®^*, similar to surfactants, which enhance the penetration of glyphosate through the plant cuticle [75]. Similar synergistic effects were previously shown in alterations of neurological behavior and morphology in *Danio reri* [76]. In our study, we treated L4 nematodes with varying concentrations of selected pesticides, and, 48 h post-treatment, we observed reduced heat-stress resistances of the animals at 37 °C. Our data show that, depending on the concentration of the pesticides, the heat stress resistance of the nematodes decreased. In the present study, for each pesticide, we selected the concentration for further experiments, which resulted in a significant decrease of survival in the Heat Stress Survival Assay (log-rank (Mantel–Cox) test; *p* value of <0.0001). The selected concentrations were 1 µM for glyphosate, 2.5 µM for fluopyram, and 5 µM for pyraclostrobin. Although LC50 values are not directly comparable between species, the selected concentration of pyraclostrobin was 1,94 mg/L, which is one hundred and twenty times the LC50 of 0.016 mg/L in *Daphnia magna* [48,77,78]. On the other site, the selected concentration of glyphosate of 169.1 µg/L was more than twenty thousand times below the LC50 of 40 mg/L in the same species. However, all three investigated pesticides showed virtually no warm-blooded animal toxicity (for example LC50 of glyphosate in rats is >5.000 mg/L; EC—Material Safety Data Sheet).

Based on the ability of nematodes to perceive their food through chemotaxis, we observed whether the pesticides and their chosen concentrations affected this elementary parameter in nematode behavior. Chemotactic attraction is controlled by neuronal mechanisms in *C. elegans* [59]. A great benefit of this observation is the well-characterized nervous system, which is functionally similar to mammals and contains 302 neurons [79]. Although we did not find any significant differences in chemotactic behavior of the nematodes after treatment with glyphosate, fluopyram, or pyraclostrobin, this might indicate the possible recovery of the nematodes post-treatment and indicates a non-neuronal effect on the nematodes as well. A similar effect was observed after treatment with either aldicarb or fenamiphos [80]. Pesticides that target the electron transport chain lead to oxidative stress and the production of reactive oxygen species (ROS) such as rotenone. The cascade of this non-direct neurologically affecting pesticide leads to the possible damage of dopaminergic neurons, and alters the attractive–repellent type behavior [81]. Additionally, it is mentioned that adult behavior is more affected by early stimulation during the larval stages [82]. In our study, we tested L4/young-adult nematodes under toxic conditions. This perhaps indicates that chemotactic alterations were recovered after 48 h treatment.

The energy metabolism is recovered after pesticide toxification. In a follow-up experiment, we examined the energy metabolism 48 h hours post-treatment with either 0.001 mM glyphosate, 2.5 µM fluopyram, or 5 µM pyraclostrobin. In all treatment groups, the ATP levels were elevated. Interestingly, the treatment with pyraclostrobin led to a significant increase of the major energy metabolite ATP. The increase in ATP concentrations could indicate the enhancement of anabolic processes, including detoxifying processes. In such cases, it is assumed that the maintenance of the energetic status of an organism was probably associated with the activation of anaerobic energy production pathways. The enhancement of anaerobic processes in response to chemical stress has also been found in other studies, e.g., in aquatic invertebrates [83,84] and in cell lines in in vitro studies [85]. Babczyńska et al. (2011) showed similar effects to those of aquatic invertebrates in which ATP levels of metal-contaminated *Agelena labyrinthica* were maintained at a relatively high level [86]. Regarding the metabolites of glycolysis, pyruvate, and lactate, we observed a nonsignificant but similar compensatory effect. Throughout the treatments, pyruvate levels were numerically but not significantly increased after 48 h of treatment. The lactate levels tended to be altered by pyraclostrobin. Through a comparison of the lactate/pyruvate ratio, none of the pesticides were found to be able to significantly alter the values. This has been previously described for lactate and pyruvate levels in rats after treatment with the organophosphorus compound malathion [87] and in eels treated with the organochlorine insecticide lindane [88]. Lactic acid concentration depends on both changes in pyruvate production and changes in cellular respiration [89]. Thus, an increase in lactic acid concentration together with a change in pyruvate may highlight the potential recovery of the nematodes after pesticide treatment. To highlight the possible recovery after pyraclostrobin toxification, we treated SH-SY5Y cells with a range of concentrations of pyraclostrobin and investigated the energy output and cell viability. The concentrations presented here have different effects. Lower concentrations (10, 100, 250, and 500 nM) only lead to a slightly significant or nonsignificant numerical decline of ATP. The higher concentrations presented here led to a significant reduction in ATP output. Regarding cell viability, for the concentrations at 1 and 5 µM, no alterations in the NADH-dependent cellular oxidoreductase-related reduction of tetrazole were visible, and we assume that the cells were intact [90]. The target of pyraclostrobin in complex III of the ETC [91] acts as an inhibitor, similar to other well-studied substances (antimycin A, stimatellin, and myxothiazol) [92]. The inhibition of complex III prevents proton transport. As a result, there is no proton gradient that is needed for the activation of complex V. ATP synthesis is blocked and declines in energy output [93,94]. Similar to our findings, a decline in energy metabolism was found in 3T3-L1 cells [95]. The treatment with 1 and 10 µM led to opposing effects on mitochondrial respiration, and therefore low concentrations may result in compensatory effects. The other complexes of the ETC may increase their activity in an attempt to overcome an incomplete inhibition of complex III [96,97,98]. These results are in line with our cell viability findings. The concentration of 5 µM rapidly decreased the energy output but did not alter the cell viability. The cells were kept alive and therefore it was possible that the organism could overcome toxification after 48 h of treatment, whereby an elevation of ATP levels was visible.

The impact of glyphosate, fluopyram, and pyraclostrobin on gene expression was also observed. To elaborate on the observed changes in lifespan and mitochondrial energy metabolism, we assessed the mRNA levels of several genes associated with longevity, healthspan, mitogenesis, and mitochondrial dysfunction. The observation of three longevity and stress-resistance associated genes, *daf-16*, *aak-2*, and *sir-2.1*, led to differential results among the pesticide treatment of the nematodes [99,100,101]. The treatment with glyphosate led to a numerical decrease of 18% in *daf-16* expression. In contrast, fluopyram reduced the expression of *sir-2.1* by 13% and did not affect *daf-16*. Both of the pesticides tended to lower the expression of *aak-2* but not significantly. Furthermore, the treatment group with pyraclostrobin revealed a drastic decrease of these three genes. Due to the observed results in the differential expression of *daf-16* and *sir-2.1,* we suggest a target-independent activation and inhibition of these genes in the presence of glyphosate and fluopyram. Furthermore, *daf-16* activates functions such as DNA binding transcription factor activity, which is activated by *sir-2.1* and *aak-2* [101]. The downregulation of *aak-2* indicates the link between energy levels and lifespan [102]. While stressors usually lead to a decrease of ATP and, therefore, affect lifespan due to the regulation of *aak-2*, we suggest that the increase of ATP leads to a negative affection of *aak-2* and reduces stress resistance. Because pyraclostrobin and fluopyram target the mitochondrial-electron transport chain [49,95] and because glyphosate was also recently described as an inhibitor of complex II [103], we observed the gene expression of an ATP synthase subunit. In this case, glyphosate treatment led to slightly decreased expression by 6%, and fluopyram treatment elevated the mRNA level by 60%. Once again, pyraclostrobin significantly decreased gene expression. Combining these results with the differential effects of the pesticides on ATP production, we suggest a possible recovery of energy metabolism despite the persistence of DNA damage after environmental stress [104,105]. Previously, it was shown in the earthworms *Eisenia fetida*, that DNA damage caused by pyraclostrobin increases over time from exposure, but reactive oxygen species decrease over time due to the activation of antioxidative enzymes such as SOD [106]. The overall significantly decreased mRNA levels in the pyraclostrobin treatment group also indicated a more nuclear effect of pyraclostrobin in comparison to its affect in the mitochondria. Previous studies revealed more persistent damage to numbers in mitochondrial DNA (mtDNA) versus nuclear DNA after treatment with ROS generators such as H_2_O_2_ in SV40-transformed fibroblasts [105]. Although *atp-2* levels were disrupted and were therefore significantly reduced, it was described earlier that damaging mitochondrial genes do not necessarily impact ATP levels in *Drosophila melanogaster* or *Caenorhabditis elegans* [107,108,109,110]. Several studies have indicated an apoptotic pathway due to the loss of mtDNA copies. As we did not observed an increase in ATP, it cannot be conclusively clarified that the treatment of these three pesticides leads to apoptosis, and it needs further investigation [111,112,113].

Finally, based on our previous findings of paraquat on mitochondrial parameters in *C. elegans* [45,87], we were interested in the possible detoxifying effects of fermented fruit, vegetable extracts and protocatechuic acid. We therefore co-treated the nematodes with the pesticides described above and with Rechtsregulat^®^Bio (RR) and protocatechuic acid (PCA), separately. RR is a manufacturer-produced pre-fermented fruit and vegetable drink containing a variety of polyphenolic metabolites [60]. Due to its health-promoting effect is a focus of nutrition today [114]. Due to the fermentation process by bacteria, not only does the composition of macromolecules change, but also the transition of bioactive compounds such as polyphenols, that change into phenolic molecules, which provides an elevated beneficial effect [61]. As a major compound of RR, protocatechuic acid (PCA) provides a well-described health benefit [115,116]. Our data show an improvement of stress resistance after co-treatment of pesticides and health-beneficial substances. Across all groups, decreased stress resistance was compensated for and raised to its initial level. Similar to these findings, we recently investigated the effect of PCA and RR after treatment with the herbicide PQ [60,117]. Both compounds improved the heat stress resistance of nematodes, due to an increase in *sir-2.1* and *daf-16* expression. Besides these genes, *skn-1* is involved in phase II detoxification processes [118]. Additionally, the key regulator of mitochondrial biogenesis, PGC1α, is absent in *C. elegans* [119]. This process is, in part, regulated via *skn-1* in nematodes [120]. In our previous study, we observed a PQ-induced increase of *skn-1*, indicating the promotion of mitochondrial turn-over [60,117]. Regarding the thermotolerance assay of the present study, we observed similar results.

## 5. Conclusions

Overall, we report the harmful effects of pesticides on non-target organisms. Although energetic processes seem to be restored after a long treatment time, other processes such as heat stress resistance were damaged permanently by the investigated pesticides. Some pesticides seem to affect gene expression in a severely damaging way, which drastically alters the expression pattern of genes. Phenolic metabolites produced by fermentation can compensate for the harmful effects of pesticides.

## Figures and Tables

**Figure 1 ijms-22-13478-f001:**
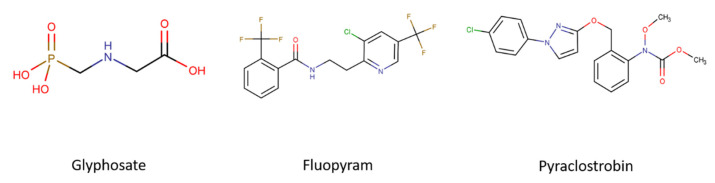
Chemical structure of the tested glyphosate, fluopyram, and pyraclostrobin. Structures were drawn with the software Chemicalize.

**Figure 2 ijms-22-13478-f002:**
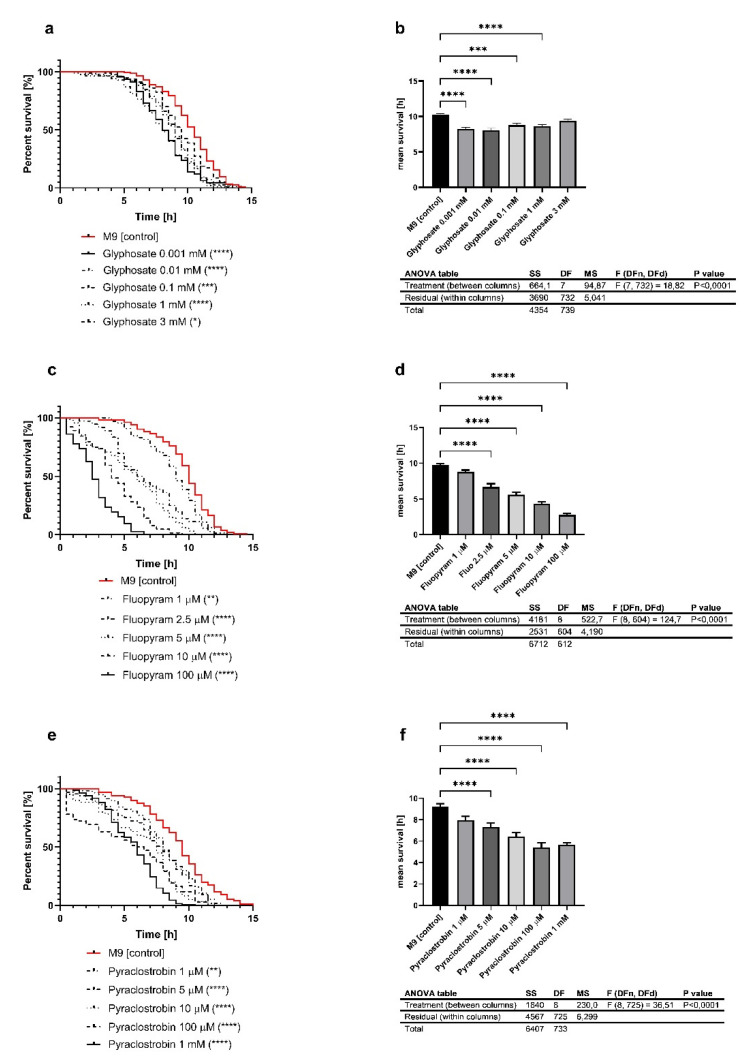
The lifespan under heat stress of *C. elegans* is reduced under exposure to glyphosate (**a**), fluopyram (**c**) and pyraclostrobin (**e**) over a range of concentrations. For heat stress experiments, the survival was assessed according to the penetration of SYTOX Green nucleic acid stain into dead cells; *n* > 60; log-rank (Mante–Cox) test; * *p* < 0.05, ** *p* < 0.01, *** *p* < 0.001 and **** *p* < 0.0001. The graphs (**b**,**d**,**f**) show in comparison the mean survival of the nematodes after pesticide treatment; *n* > 61; mean ± SEM; one-way ANOVA with Tukey’s comparison post hoc test; *** < 0.001 and **** < 0.0001.

**Figure 3 ijms-22-13478-f003:**
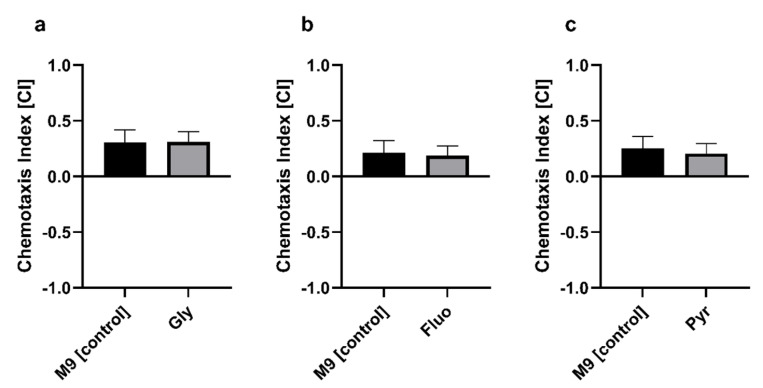
Chemotaxis index after exposure to glyphosate (**a**), fluopyram (**b**), and pyraclostrobin (**c**). The nematodes were treated with the pesticides for 48 h before the experiment; *n* = 9; mean ± SEM; Student’s t test; no significance.

**Figure 4 ijms-22-13478-f004:**
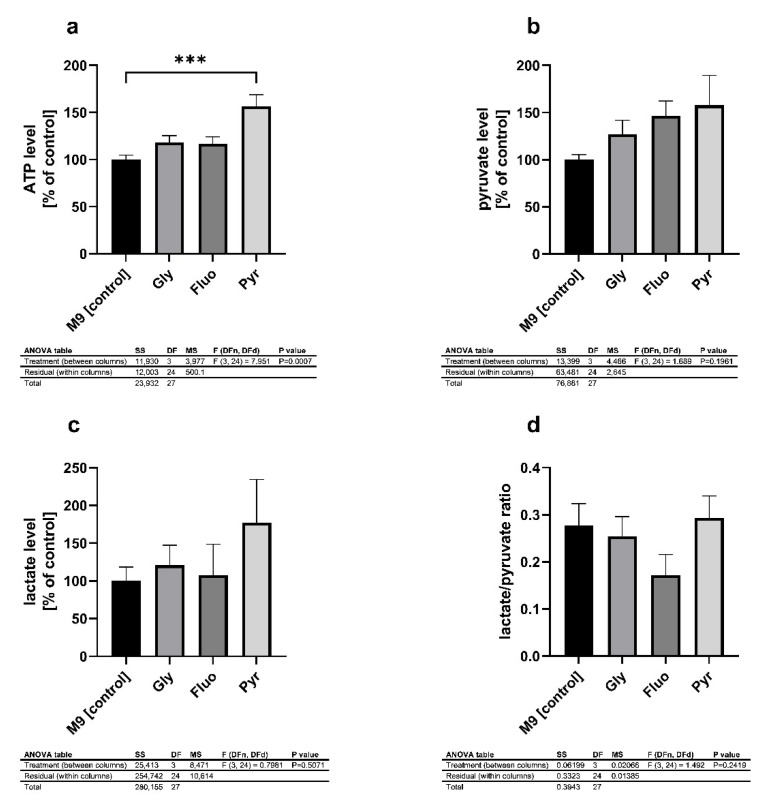
Determination of intracellular ATP levels (**a**), pyruvate levels (**b**), lactate levels (**c**) and the lactate/pyruvate ratio (**d**) of wild-type nematodes after exposure to glyphosate (0.001 mM), fluopyram (2.5 µM) and pyraclostrobin (5 µM). ATP levels were measured using the ATPlite luminescence assay. Lactate and pyruvate levels were assessed by using a colorimetric assay kit. The values were normalized to protein concentrations and percent of the control group treated with M9 buffer; *n* = 8; mean ± SEM; one-way ANOVA with Tukey’s comparison post hoc test; *** *p* < 0.001.

**Figure 5 ijms-22-13478-f005:**
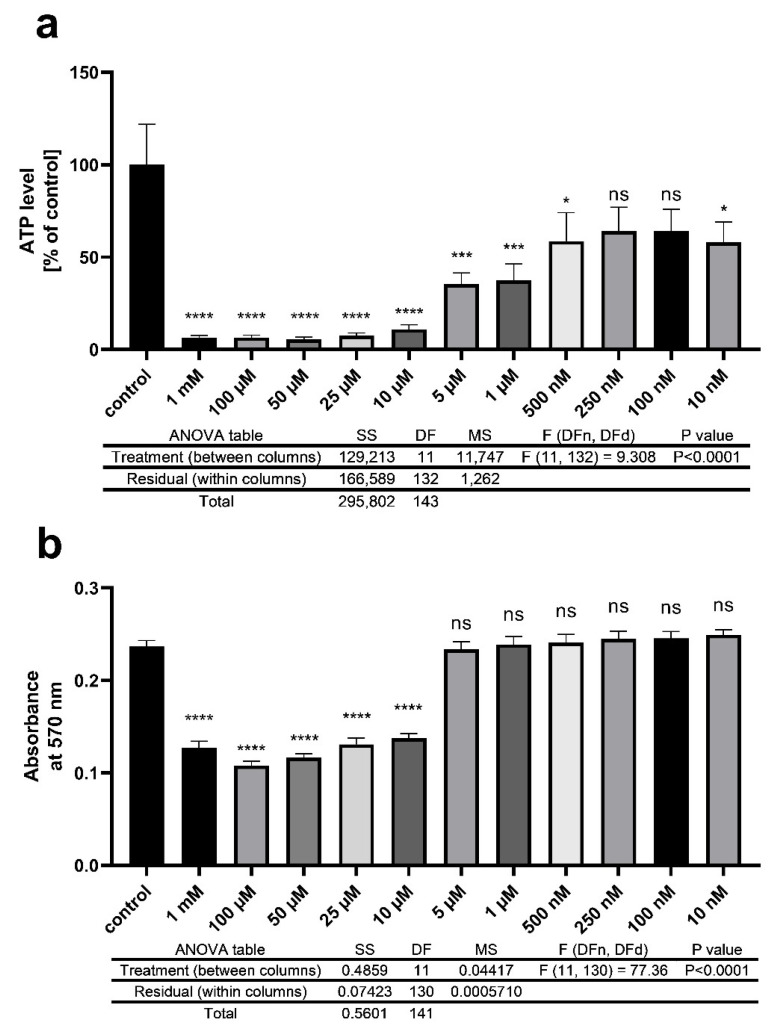
Determination of intracellular ATP levels (**a**) and cell viability (**b**) of SH-SY5Y cells after exposure to pyraclostrobin (5 µM) for 24 h. ATP levels were measured using the ATPlite luminescence assay. Lactate and pyruvate levels were assessed by using a colorimetric MTT assay. The values were normalized to 10,000 cells and percent of the control group treated with 0.1% EtOH in medium; *n* = 12; mean ± SEM; one-way ANOVA with Tukey’s comparison post hoc test; * *p* < 0.05, *** *p* < 0.001 and **** *p* < 0.0001.

**Figure 6 ijms-22-13478-f006:**
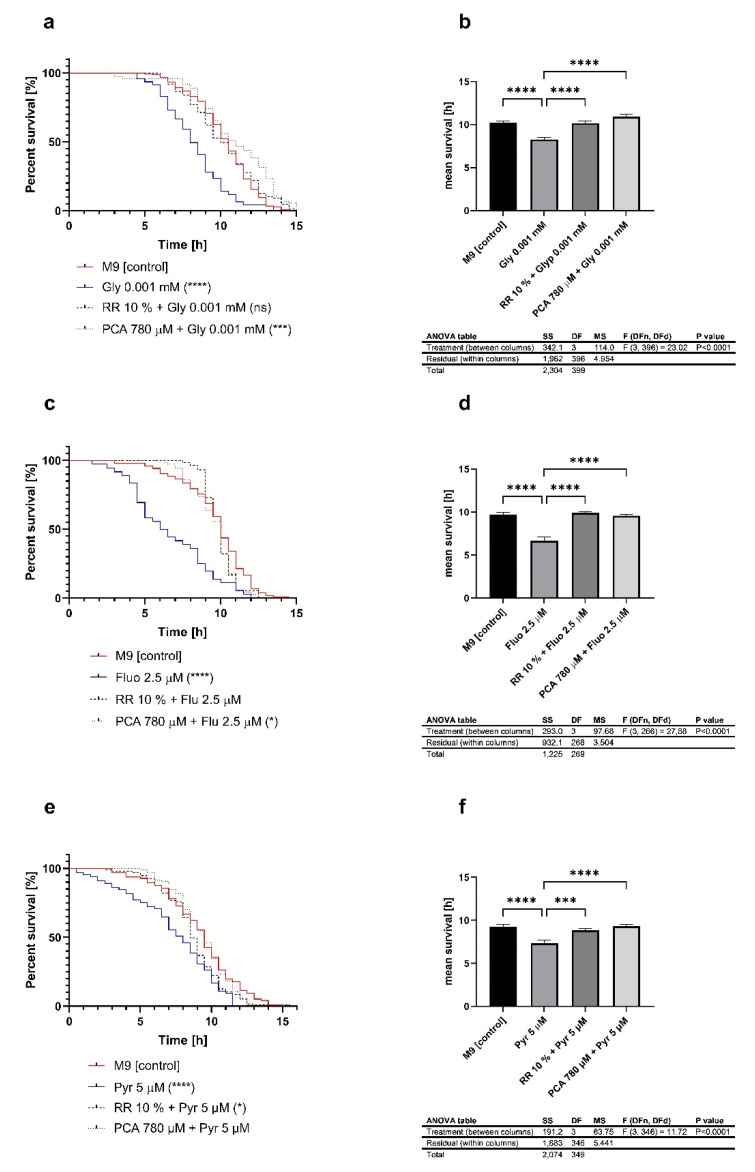
Heat stress resistance of wild-type *C. elegans* at 37 °C after exposure to glyphosate 0.001 mM, fluopyram 2.5 µM, pyraclostrobin 5 µM and in combination with either RR 10% or PCA 780 µM (**a**–**f**). For heat stress experiments, the survival was assessed according to the penetration of SYTOX Green nucleic acid stain into dead cells; *n* > 60; log-rank (Mantel-Cox) test; * *p* < 0.05, *** *p* < 0.001 and **** *p* < 0.0001. The graphs (**b**,**d**,**f**) show in comparison the mean survival of the nematodes after pesticide treatment; *n* > 60; mean ± SEM; one-way ANOVA with Tukey’s comparison post hoc test; *** *p* < 0.001 and **** *p* < 0.0001.

**Table 2 ijms-22-13478-t002:** Oligonucleotide primer sequences and product sizes for quantitative real-time PCR. Concentration was 0.1 µM for all primers.

Primer	Sequence	Product Size (bp)
*aak-2* *NM_001029697.6*	5′-tgcttcaccatatgctctgc-3′ 5′-gtggatcatctcccagcaat-3′	219
*ama-1* *NM_068122.9*	5′-ccaggaacttcggctcagta-3′ 5′-tgtatgatggtgaagctggcg-3′	85
*act-2* *NM_001383398.2*	5′-cccactcaatccaaaggcta-3′ 5′-gggactgtgtgggtaacacc-3′	168
*atfs-1* *NM_074114.7*	5′-tcggcgatcgatcagctaac-3′ 5′-agaatcagttcttggattagggga-3′	75
*atp-2* *NM_065710.8*	5′-tccaagtcgctgaggtgttc-3′ 5′-aggtggtcgagttctcctga-3′	151
*daf-16* *NM_001026422.6*	5′-tcctcattcactcccgattc-3′ 5′-ccggtgtattcatgaacgtg-3′	175
*sir-2.1* *NM_001268555.5*	5′-tggctgacgattcgatggat-3′ 5′-atgagcagaaatcgcgacac-3′	179
*skn-1* *NM_171345.6*	5′-acagggtggaaaaagcaagg-3′ 5′-caggccaaacgccaatgac-3′	246

bp, base pairs; *aak-2*, AMP-activated kinase; *ama-1*, amanitin resistant; *act-2*, actin; *atfs-1*, activating transcription factor associated with stress; *atp-2*, ATP synthase subunit; *daf-16*, abnormal dauer formation; *sir-2.1*, yeast sir related, *skn-1*, skinhead.

**Table 3 ijms-22-13478-t003:** Relative normalized mRNA expression levels of wild-type nematodes treated with 0.001 mM glyphosate, 2.5 µM fluopyram, or 5 µM pyraclostrobin. mRNA expression of M9 (control) is 100%. *n* = 6–8; mean ± SEM; Student’s t test; * *p* < 0.05, ** *p* < 0.01, *** *p* < 0.001 and **** *p* < 0.0001.

	M9 (Control)	Gly 0.001 mM	M9 (Control)	Fluo 2.5 µM	M9 (Control)	Pyr 5 µM
*daf-16*	100.0 ± 11.34	81.37 ± 6.095	100.0 ± 5.940	103.2 ± 11.74	100.0 ± 2.323	34.42 ± 4.417 **** *p* < 0.0001
*sir-2.1*	100.0 ± 5.070	114.9 ± 5.327	100.0 ± 6.891	86.87 ± 4.198	100.0 ± 16.06	28.68 ± 13.85 ** *p* = 0.0072
*aak-2*	100.0 ± 15.92	91.36 ± 8.859	100.0 ± 12.00	86.56 ± 11.17	100.0 ± 13.20	20.20 ± 6.315 *** *p* < 0.0007
*atp-2*	100.0 ± 6.574	93.64 ± 4.589	100.0 ± 12.47	160.5 ± 19.61 * *p* = 0.0263	100.0 ± 9.440	13.54 ± 3.302 **** *p* < 0.0001
*skn-1*	100.0 ± 15.80	74.41 ± 7.508	100.0 ± 12.92	95.47 ± 7.471	100.0 ± 7.316	14.72 ± 5.783 **** *p* < 0.0001
*atfs-1*	100.0 ± 13.29	68.17 ± 6.165	100.0 ± 8.947	83.77 ± 5.508	100.0 ± 6.257	39.34 ± 9.921 *** *p* < 0.0004

## Data Availability

The dataset generated during this study is available from the corresponding author on reasonable request.

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
