# Peer review of "Effects of Pesticides on Longevity and Bioenergetics in Invertebrates—The Impact of Polyphenolic Metabolites"

_ijms, 2021, doi:10.3390/ijms222413478_

Round 1

Reviewer 1 Report

  1. This is an interesting study to evaluate the effect of pesticides on non-target organisms and focus on mitochondria. Although the results revealed the effects on mitochondrial level, several questions are suggested to state clearly in the manuscript.
  2. Generally there are pesticides safety directorates of different pesticides; the author should discuss the concentrations of glyphosate, fluopyram and pyraclostrobin that used in this study with the pesticides safety directorates.
  3. How about the change of mitochondria DNA copy number in pesticides treated nematodes, did the pesticides induce a mitochondria dependent apoptosis?
  4. How to distinguish the toxicity of pesticides to nematodes is mitochondrial effect but not interfere neuronal processes?
  5. In the part of discussing the possible detoxifying effects of pesticides, author concluded the phase II detoxification processes. Are the detoxifying effects correlated with the actions of the pesticides?

6. From this manuscript, I could feel the enthusiasm of the author on this study! The above questions are suggested to make the study more convincing; I think the author dose not need to do other experiments, just need to make statement in appropriate paragraph. 

Author Response

Reviewer 1

  1. This is an interesting study to evaluate the effect of pesticides on non-target organisms and focus on mitochondria. Although the results revealed the effects on mitochondrial level, several questions are suggested to state clearly in the manuscript.
  2. Generally there are pesticides safety directorates of different pesticides; the author should discuss the concentrations of glyphosate, fluopyram and pyraclostrobin that used in this study with the pesticides safety directorates.
    • Thank you for your point. In the present study, for each pesticide, we selected the concentration for further experiments, which resulted in a significant decrease of survival in the Heat stress Survival Assay (log-rank (Mantel–Cox) test; p value of < 0.0001). The selected concentrations were 1 µM for glyphosate, 2.5 µM for fluopyram, and 5 µM for pyraclostrobin. Although LC50 values are not directly comparable between species, the selected con-centration of pyraclostrobin was 1.94 mg/l, which is one hundred and twenty times above the LC50 of 0.016 mg/l in Daphnia magna. On the other site, the selected concentration of glyphosate of 169.1 µg/l was more than twenty thousand times below the LC50 of 40 mg/l in the same species. However, all three investigated pesticides showed virtually no warm-blooded animal toxicity (for example LC50 of glyphosate in rats is >5.000 mg/l; EC—Material Safety Data Sheet).
    • This point was integrated into the manuscript (lines 360 to 370).
  3. How about the change of mitochondria DNA copy number in pesticides treatednematodes, did the pesticides induce a mitochondria dependent apoptosis?
    • These are very interesting details. Unfortunately, it has not been investigated in this study. Previously, it was shown in the earthworms Eisenia fetida that DNA damage caused by pyraclostrobin increases over time from exposure, but reactive oxygen species decrease over time due to the activation of antioxidative enzymes such as SOD. The overall significantly decreased mRNA amounts in the pyraclostrobin treatment group also indicated a more nuclear effect of pyraclostrobin versus in the mitochondria. Previous studies revealed more persistent damage to numbers in mitochondrial DNA (mtDNA) versus nuclear DNA after treatment with ROS generators such as H2O2 in SV40-transformed fibroblasts. Although atp-2 levels were disrupted and were therefore significantly reduced, it was described earlier that damaging mitochondrial genes do not necessarily impact ATP levels in Drosophila melanogaster or Caenorhabditis elegans. Several studies have indicated an apoptotic pathway due to the loss of mtDNA copies. Unless we observed an increase in ATP, it cannot be conclusively clarified that the treatment of these three pesticides leads to apoptosis, and it needs further investigation.
    • The important point has been discussed now (lines 454 to 467)
  4. How to distinguish the toxicity of pesticides tonematodes is mitochondrial effect but not interfere neuronal processes?
    • Based on the ability of nematodes to perceive their food through chemotaxis, we observed whether the pesticides and their chosen concentrations affected this elemen-tary parameter in nematode behavior. Chemotactic attraction is controlled by neuronal mechanisms in elegans. A great benefit of this observation is the well-characterized nervous system, which is functionally similar to mammals and contains 302 neurons. Although we did not find any significant differences in chemotactic behavior of the nematodes after treatment with glyphosate, fluopyram, or pyraclostrobin, this might indicate a possible recovery of the nematodes post-treatment and indicates a non-neuronal effect on the nematodes as well.
    • This point has now been discussed (lines 368 to 374)
  5. In the part of discussing the possible detoxifying effects of pesticides, author concluded the phase II detoxification processes. Are the detoxifying effects correlated with the actions of the pesticides?
    • We haven’t tested detoxifying enzymes but we recently investigated the effect of protocatechuic acid and Rechtsregulat®Bio after treatment with the herbicide paraquat (PQ). Both compounds improved the heat-stress resistance of the nematodes which is based on the increase in sir-2.1 and daf-16 Besides these genes also an influence of skn-1, which is involved in phase II de-toxification processes. Additional, the key regulator of mitochondrial biogenesis, PGC1α, is absent in C. elegans. This process is in part regulated via skn-1 in nematodes. In our previous study we observed an increase of skn-1 and indicate a pro-motion of mitochondrial turnover. Regarding the thermotolerance assay of the present study we can observe identical mechanisms.
    • This point has also been added to the discussion (lines 481 to 489)
  6. From this manuscript, I could feel the enthusiasm of the author on this study! The above questions are suggested to make the study more convincing; I think the author dose not need to do other experiments, just need to make statement in appropriate paragraph. 
    • We thank the reviewer for this comment. Indeed we are excited by the them and loved to investigate the effects of pesticides in the model organism elegans.

Reviewer 2 Report

The manuscript of Schmitt and Eckert describes the effects of some pesticides on Caenorhabditis elegans. The manuscript is well written, the methods are adequate and the results support the conclusions made. I have only minor points:

1) line 13: please omit "like"

2) lines 108-11: I am not perfectly sure if the concentration of ethanol was the same in the experimental and control groups. Please clarify.

3) lines 156-159: Why two colorimetric tests were used?

4) lines 104-105: It is sufficient to give that p<0.05 was considered to be statistically significant.

5) Figure 2: Please unify the rank of pesticides concentration (from lowest to highest).

6) line 218 and elsewhere: please provide the whole ANOVA results (F, df, etc.)

7) line 283: he should be read it

8) lines 364-365: If the statistical result is non-significant then it is not possible to talk about numerical difference. 

Author Response

Reviewer 2

The manuscript of Schmitt and Eckert describes the effects of some pesticides on Caenorhabditis elegans. The manuscript is well written, the methods are adequate and the results support the conclusions made. I have only minor points:

1) line 13: please omit "like"

2) lines 108-11: I am not perfectly sure if the concentration of ethanol was the same in the experimental and control groups. Please clarify.

3) lines 156-159: Why two colorimetric tests were used?

4) lines 104-105: It is sufficient to give that p<0.05 was considered to be statistically significant.

5) Figure 2: Please unify the rank of pesticides concentration (from lowest to highest).

6) line 218 and elsewhere: please provide the whole ANOVA results (F, df, etc.)

7) line 283: he should be read it

8) lines 364-365: If the statistical result is non-significant then it is not possible to talk about numerical difference. 

-           We thank the reviewer for the kind revision. All minor points were corrected.

Reviewer 3 Report

The authors of the manuscript "EFFECTS OF PESTICIDES ON LONGEVITY AND BIOENERGETICS IN INVERTEBRATES – IMPACT OF POLYPHENOLIC METABOLITES" conducted a study to evaluate the effects of three common pesticides on C. elegans and of the potential therapeutic effect of polyphenolic metabolites. 

Major points:

  1. The authors justified the choice of the concentrations of fluopyram and pyraclostrobin, based on the effect on C.elegans survival. For glyphosate  the  concentration was chose because it led to a decrease of 19 % in mean lifespan. Can the authors provide a more compelling explanation?
  2. The study did not provide a detailed study of the effects of pesticides on the different systems of C.elegans. It would be beneficial for the manuscript to include an histological/cell study of the metabolites considered in the study.
  3. The authors observed an increase in ATP in nematodes treated with pyraclostrobin. At the same time, the expression of atp-2 was reported to be increased. Can the authors explain this discrepancy?

Minor point: The manuscript includes multiple sentences that do not make sense. Ex. line 37-38: "Not only to reduce crop losses by pests and thus can improve quality and yield of the produce". Line 74-75 "Even pesticides should only act on plants, fungi, or insects, today's evidence grows that they may also act on mammalian mitochondria..."

Author Response

Reviewer 3

(x) Extensive editing of English language and style required 

MPDI language check has been applied

Major points:

1. The authors justified the choice of the concentrations of fluopyram and pyraclostrobin, based on the effect on C.elegans survival. For glyphosate  the  concentration was chose because it led to a decrease of 19 % in mean lifespan. Can the authors provide a more compelling explanation?

  • Thank you for your point. In the present study, for each pesticide, we selected the concentration for further experiments, which resulted in a significant decrease of survival in the Heat stress Survival Assay (log-rank (Mantel–Cox) test; p value of < 0.0001). The selected concentrations were 1 µM for glyphosate, 2.5 µM for fluopyram, and 5 µM for pyraclostrobin. Although LC50 values are not directly comparable between species, the selected con-centration of pyraclostrobin was 1.94 mg/l, which is one hundred and twenty times above the LC50 of 0.016 mg/l in Daphnia magna. On the other site, the selected concentration of glyphosate of 169.1 µg/l was more than twenty thousand times below the LC50 of 40 mg/l in the same species. However, all three investigated pesticides showed virtually no warm-blooded animal toxicity (for example LC50 of glyphosate in rats is >5.000 mg/l; EC—Material Safety Data Sheet).
  • This point was integrated into the manuscript (lines 360 to 370).

2. The study did not provide a detailed study of the effects of pesticides on the different systems of C.elegans. It would be beneficial for the manuscript to include an histological/cell study of the metabolites considered in the study.

  • elegans was chosen since it provides a simple but robust animal model to study mitochondrial function in a whole organism (Dilgerger B. et al. (2020) Aging 12(12): 12268-12284). However, we agree with the reviewer, that the benefit of our study represents also a limitation, since we can not extrapolate the results to the cellular level. Thus, based on the suggestion of the reviewer we now provide new data investigating the effects of pyraclostrobin on key mitochondrial mechanisms in neuroblastoma cells, which were chosen as neuronal model. To highlight the possible recovery after a pyraclostrobin toxification we treated SH-SY5Y- cells with a range of concentrations of pyraclostrobin and investigated the energy output and the cell viability. The concentrations presented here have different effects. Lower concentrations (10 nM, 100 nM, 250 nM, 500 nM) only lead to slightly significant or non-significant numerical decline of ATP. The here presented higher concentrations lead to a significant reduction in ATP output. Regarding the cell viability, for the concentrations 1 µM and 5 µM no alterations in the NADH-dependent cellular oxidoreductase related reduction of tetrazole are visible and we assume that the cells are intact. The target of pyraclostrobin is complex III of the ETC and acts inhibitory on it like other well-studied substances (antimycin A, stimatellin, and myxothiazol) [91]. The inhibition of complex III prevents proton transport. As a result, there is no proton gradient which is needed for the activation of complex V. The ATP synthesis is blocked and it comes to a decline in energy output. Similar to our findings a decline in energy metabolism has been found in 3T3-L1 cells. Interestingly, the treatment with 1 µM and 10 µM led to opposing effects on mitochondrial respiration and therefore low concentrations may result in compensatory effects. The other complexes of the ETC may increase their activity in an attempt to overcome an incomplete inhibition of the complex III. These results are in line with our cell viability findings. The concentration of 5 µM rapidly decreases the energy output but doesn’t alter the cell viability. The cells are kept alive and therefore it is possible that the organism can overcome the toxification after 48 h of treatment and an elevation of ATP levels is visible.
  • We add the new data (figure 5; results lines 268 to 288) and discussed the results (lines 407 to 429).

3. The authors observed an increase in ATP in nematodes treated with pyraclostrobin. At the same time, the expression of atp-2 was reported to be increased. Can the authors explain this discrepancy?

  • We agree with the reviewer that the data on ATP levels and gene expression of atp-2 that represents seem to be in contrast. However, previously, it was shown in the earthworms Eisenia fetida that DNA damage caused by pyraclostrobin increases over time from exposure, but reactive oxygen species decrease over time due to the activation of antioxidative enzymes such as SOD. The overall significantly decreased mRNA amounts in the pyraclostrobin treatment group also indicated a more nuclear effect of pyraclostrobin versus in the mitochondria. Previous studies revealed more persistent damage to numbers in mitochondrial DNA (mtDNA) versus nuclear DNA after treatment with ROS generators such as H2O2 in SV40-transformed fibroblasts. Although atp-2 levels were disrupted and were therefore significantly reduced, it was described earlier that damaging mitochondrial genes do not necessarily impact ATP levels in Drosophila melanogaster or Caenorhabditis elegans. Several studies have indicated an apoptotic pathway due to the loss of mtDNA copies. Unless we observed an increase in ATP, it cannot be conclusively clarified that the treatment of these three pesticides leads to apoptosis, and it needs further investigation.
  • The important point has been discussed now (lines 454 to 467)

Minor point: The manuscript includes multiple sentences that do not make sense. Ex. line 37-38: "Not only to reduce crop losses by pests and thus can improve quality and yield of the produce". Line 74-75 "Even pesticides should only act on plants, fungi, or insects, today's evidence grows that they may also act on mammalian mitochondria..."

  • Minor points has been corrected.

Round 2

Reviewer 3 Report

Dear Authors,

I appreciate the efforts made to improve the article. The quality and the grammar of the manuscript was improved, and the new experiments required made the work more compelling.